# Seismic Imaging of Mineral Exploration Targets: Evaluation of Ray- vs. Wave-Equation-Based Pre-Stack Depth Migrations for Crooked 2D Profiles

**Brij Singh** [1,*] and **Michał Malinowski** [1,2]

1   Institute of Geophysics, Polish Academy of Sciences Warsaw, 01-452 Warsaw, Poland
2   Geological Survey of Finland, FI-02151 Espoo, Finland
*   Correspondence: bsingh@igf.edu.pl

**Abstract:** Seismic imaging is now a well-established method in mineral exploration with many successful case studies. Seismic data are usually imaged in the time domain (post-stack or pre-stack time migration), but recently pre-stack depth imaging has shown clear advantages for irregular/sparse acquisitions and very complex targets. Here, we evaluate the effectiveness of both ray-based and wave-equation-based pre-stack depth imaging methodologies applied to crooked-line 2D seismic reflection profiles. Seismic data were acquired in the Kylylahti mining area in eastern Finland over severely folded, faulted and subvertical Kylylahti structure, and associated mineralization. We performed 3D ray-based imaging, i.e., industry-standard Kirchhoff migration and its improved version (coherency migration, CM), and wave-equation-based migration, i.e., reverse time migration (RTM) using a velocity model built from first-arrival traveltime tomography. Upon comparing the three different migrations against available geological data and models, it appeared that CM provided the least noisy and well-focused image, but failed to image the internal reflectivity of the Kylylahti formation. RTM was the only method that produced geologically plausible reflections inside the Kylylahti formation including a direct image of the previously known shallow massive sulfide mineralization.

**Keywords:** Hardrock seismics; mineral exploration; depth imaging; coherency migration; reverse-time migration

## 1. Introduction

Seismic reflection surveying since its conceptualization nearly three decades ago has become a popular choice for deep mineral targeting [1,2]. By and large, most of the seismic surveys conducted in the hardrock environment are 3D surveys targeting deep mineral deposits of mainly volcanogenic origin (e.g., [3–12] and references therein). Three-dimensional surveys are essential for accurate target delineation but are otherwise financially expensive, involve huge logistical costs, are labor intensive and require larger computational resources, which significantly increases the processing time. Conversely, 2D seismic profiles are a great means of affordable regional assessment in hardrock seismic exploration and this approach has been successfully applied in several projects (e.g., [13–15]). Two-dimensional surveys can also be used for initial seismic assessment of the area and can form a basis for the development of a full-scale 3D survey (e.g., [16,17]).

The most common approach to processing hardrock seismic data is to compensate first for the common depth point (CDP) dispersal due to the dipping strata and enhance diffractions through the dip move-out (DMO). This is followed by post-stack time-domain migration (DMO-PoSTM) which produces the desired results for targets that are well illuminated and where the geology is not too complex (i.e., dips are moderate), with prominent impedance contrasts between the rock types, as well as for the data characterized by good signal-to-noise ratio. Motivated by several successful case studies, reflection seismics is getting more ambitious by targeting low seismic impedance contrasts placed

in geologically complex settings. Moreover, terrain access, noisy mining infrastructure and logistics cause many seismic surveys to result in irregular/sparse 3D or crooked-line 2D coverage, which in turn brings imperfect illumination of the target. Recent case studies performed in such conditions suggest that pre-stack depth migration (PreSDM) can effectively deal with such challenges where the conventional time-domain approach fails [18–22].

PreSDM is routinely applied in imaging oil and gas exploration targets. PreSDM methods are generally classified as ray-based and wave-equation-based migrations. The efficacy of both of these migration categories has already been thoroughly tested, and it is well established for over two decades now that wave-equation-based migrations provide superior imaging of the complex subsurface (e.g., subsalt imaging). However, in the context of mineral exploration, PreSDM application is very limited. The application of ray-based methods is so far concentrated on Kirchhoff pre-stack depth migration (KPreSDM) and its advanced forms i.e., coherency migration (CM) [20,23], and Fresnel Volume Migration (FVM)—a refined version of beam-type migration [18,19,21,22]. For wave-equation-based methods, one-way wave-equation migration was successfully used both for imaging and velocity model building [24]. Recently, two-way wave-equation reverse-time migration (RTM) was also applied to both 2D and 3D datasets for the delineation of iron-ore deposits in Ludvika, Sweden [25,26]. Each of these methods varies in terms of imaging principle, sensitivity to velocities, computational requirement and ability to handle the type of arrivals (multipathing). So far, they had been tested independently, or in combination (in a few cases) on different datasets, meaning different processing workflow was followed. In addition, the velocity models used for subsequent depth migration were built using different strategies. Therefore a comprehensive comparison of different PreSDM methods in the mineral exploration context is unavailable for review. In the present study, we showcase a comparison of results obtained from both types of depth migration methods. Here, we used data from two 2D crooked seismic profiles acquired in the Kylylahti mining area in eastern Finland. We test the effectiveness of both ray-based PreSDM methods (KPreSDM and CM), and two-way wave-equation-based migration (RTM). We use the same pre-processed data and velocity model derived from first-arrival traveltime tomography (FAT).

Our paper is organized as follows. In Section 2, we provide an overview of the associated geology of the area, data acquisition and processing applied to the data. Then we describe the key aspects of the velocity model building in hardrock environment, and, finally, we provide the theoretical background for KPreSDM, CM and RTM. In Section 3, we showcase the obtained velocity model using FAT followed by subsequent depth images obtained from KPreSDM, CM and RTM. Finally, we validate all migrated results against available geological information (drillholes, mapped fault surfaces and a geological common earth model, CEM).

## 2. Materials and Methods

### 2.1. Geological Background

The Outokumpu mining district situated in eastern Finland has been famous for copper mining since the early nineteenth century during which the Outokumpu area hosted several mines. Our direct area of interest is the Kylylahti polymetallic (Cu-Co-Zn-Ni-Ag-Au) semi-massive to massive sulfide (S/MS) deposit. The Outokumpu ore belt comprises of ophiolitic slices of upper mantle rocks originating from oceanic lithosphere and paleo-Proterozoic turbiditic deep-water sediments. Outokumpu nappe was thrusted over the Archean basement and was eventually strongly deformed (Figure 1). Over time, the depleted upper mantle rocks underwent metamorphic alteration and transformed into Outokumpu assemblage serpentinite-carbonate-skarn-quartz hosting the mineralization [27].

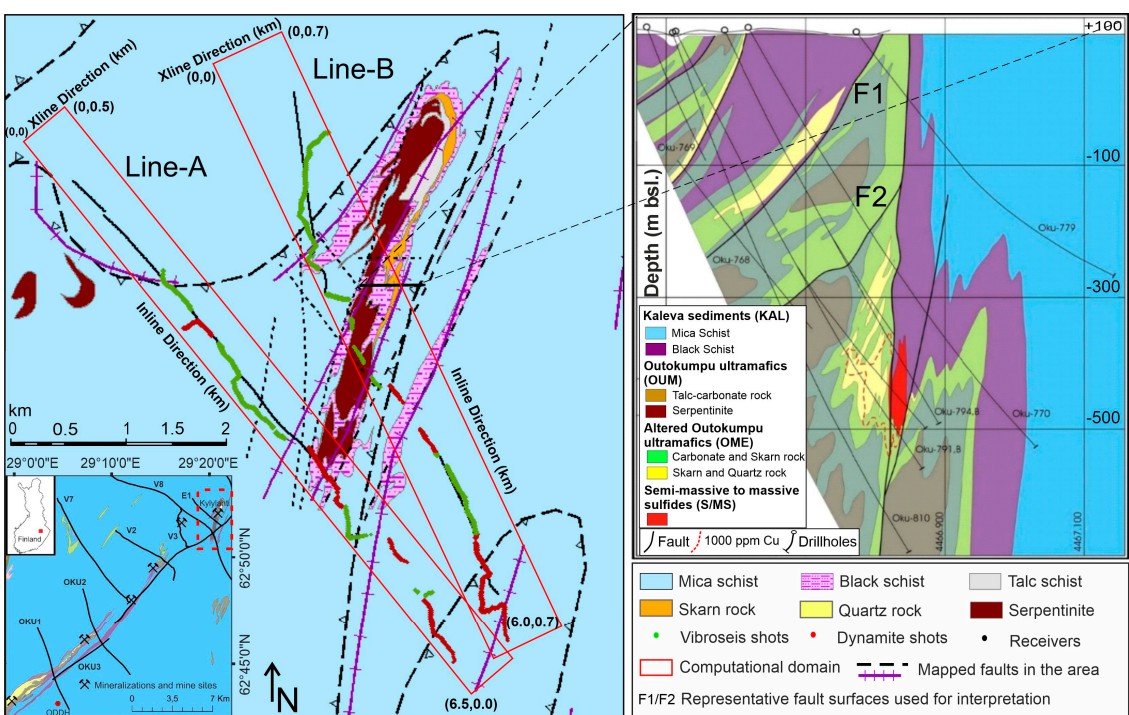

**Figure 1.** Acquisition geometry of COGITO-MIN Line-A and Line-B is shown on top of the geological map of the Kylylahti area (left panel). Red boxes show the computational domain inside which migrations are performed for both lines (6.5 × 0.5 km for Line-A, and 6.0 × 0.7 km for Line-B). Brown dots show dynamite shots, green dots show Vibroseis points and black dots mark receivers. The simplified geological map of the Outokumpu ore belt is shown in the inset (left panel). Black lines in the inset mark previously acquired 2D seismic profiles and the red dotted rectangle shows the Kylylahti area. A representative cross-section through the Kylylahti formation is shown in the top-right panel (modified after [27]).

The overall geology of the Kylylahti area can be divided into four main units (see top-right panel, Figure 1); (a) Outokumpu ultramafics (OUM), mainly consisting of serpentinite and carbonate rocks, hosting (b) semi- or massive sulfide mineralization (S/MS) and surrounded by the (c) altered Outokumpu ultramafics (OME) composed of carbonate-skarn-quartz rocks, and (d) Kalevian metasediments (black schists and turbiditic mica schists). The OUM and OME together forms a nearly N–S trending narrow, sub-vertical formation known as the Outokumpu assemblage rocks (herein referred to as Kylylahti formation). The Kylylahti formation shows strong foliation within a tight synformal fold structure with the mineralization located along a near-vertical eastern limb. Therefore, the structural complexity of the Kylylahti formation constitutes a very challenging target for surface seismic methods.

For a better assessment of the associated geophysical properties in the area, petrophysical studies were carried out on the most common rocks found in Kylylahti [28]. A significantly higher average density is measured for Kylylahti S/MS mineralization (3.7 gm/cm$^3$), followed by Outokumpu assemblage rocks (3.0 gm/cm$^3$) and for Kalevian sediments with a bit under 2.8 gm/cm$^3$. The average P-wave velocity for Outokumpu assemblage rocks was found to be a little above 6 km/s, whereas low velocities of ~5.5 km/s are measured for black schist and mica schist. Therefore, the contact between sulfide mineralization and the host rock should produce a detectable signal at the surface attributed mainly to its higher density. In addition, Outokumpu assemblage rocks have a clear contrast with the mica schist and black schist. We should also expect some reflectivity due to the alteration within the Outokumpu assemblage e.g., talc-carbonate rocks exhibit significantly lower P-wave velocities than the serpentinites.

## 2.2. COGITO-MIN 2D Seismic Survey

Two crooked 2D seismic reflection profiles (namely Line-A, and Line-B) were acquired in the Kylylahti mine area in 2016 within the COGITO-MIN project (COst-effective Geophysical Imaging Techniques for supporting Ongoing MINeral exploration in Europe) [29]. The two profiles, each ca. 6 km long, were oriented approximately NNW–SSE, cross-cutting obliquely the strike of the serpentinized Kylylahti rocks (Figure 1). The survey consisted of both dynamite and Vibroseis sources. Vibroseis data were acquired using two 9.5-t Vibroseis trucks (INOVA UniVib). Data were recorded using a Wireless Seismic RT2 nodal system equipped with a single 10 Hz geophone per receiver station. Acquisition was challenging due to the dense forest, open fields, swamps and mining infrastructure. The crookedness and gaps in both profiles were mainly attributed to these factors. All parameters related to data acquisition are summarized in Table 1.

**Table 1.** Acquisition parameters of the COGITO-MIN 2D profiles.

| Data Acquisition | Line-A | Line-B |
|---|---|---|
| Receiver Spacing (m) | 10 | |
| Source Spacing (m) | 20 | |
| Vibroseis Sweep (Hz) | 4–220 | |
| Number of Sweeps/Shot Point | 3 | |
| Dynamite Charge Size (g) | 120–240 g | |
| Shot Hole Depth (m) | 1.5–2.5 | |
| Channels | 577 | 574 |
| Vibroseis Source Points | 121 | 152 |
| Dynamite Source Points | 98 | 85 |

## 2.3. Data Pre-Processing

The pre-processing applied to the data before migration followed the same workflow as presented in [19] for processing the COGITO-MIN 2D data (Table 2). There was, however, a notable difference in how the refraction statics was handled. Our primary concern was to have a statics solution compatible with the velocity model used for migration. Earlier, [22] advocated for correcting the first-break picks with the refraction statics when building the velocity model using FAT. Here we adopted a simpler approach in which the data are only corrected for the short-wavelength (residual) part of refraction statics. The long-wavelength part is handled through wave propagation in the velocity model.

## 2.4. Velocity Model Building in Hardrock Environment

Since coherent reflections are generally absent in the Kylylahti data, conventional techniques of velocity model building such as ray-based reflection tomography are not applicable. As an alternative, FAT has been successfully applied for the depth migration of hardrock seismic data [9,18,22]. The inherent limitation of the FAT is its low resolution (on the order of ten times the dominant wavelength), which makes it extremely difficult to resolve the weathered layer–bedrock contact. Moreover, the low velocity gradient present in hardrock environment also severely limits the depth penetration of refracted arrivals.

Alternatively, [18] proposed an integrative approach of combining the tomographic near-surface velocity model by incorporating drilling information and utilizing residual moveout-based Migration Velocity Analysis (MVA) [30] to build a velocity model for depth imaging. However, this methodology is only suitable for cases where a strong coherent reflectivity from the target area is observed, as well as no strong lateral velocity variation is present. The MVA-type approach was also applied by [31] based on gathers extracted from one-way wave-equation migration. Recently, we also tested the applicability of early-arrival full-waveform inversion (FWI) to build a high-resolution velocity model for subsequent imaging (including RTM) [26,32]. One of the conclusions from our FWI trial is that the overall computational expense vs. uplift in imaging is too high. A reasonably well-focused

depth image (also from RTM) can be obtained using a low-resolution FAT model ([26]). Therefore, in the case of the current study, we used a simpler approach (i.e., FAT).

**Table 2.** Processing steps for Line-A and Line-B.

| Process | Parameter |
|---|---|
| Data Read | 3.0 s SEG-Y Data |
| Match Filter | Match Dynamite to Vibroseis |
| Geometry Setup * | Crooked Line; 5 m CDP Spacing |
| Refraction Statics ** | 2-Layer Model |
| Elevation statics | Floating datum |
| Spherical Divergence Correction | $v^2t$ Function |
| Bandpass Filter | 42-48-200-220 Hz |
| Automatic Gain Control (AGC) | 250 ms |
| Deconvolution | Spiking 150 ms/Predictive 150/12 ms |
| Airwave Mute | 330 m/s |
| Coherency Filter | F-X Decon., 15 Traces—100 ms Window |
| Linear Noise Removal | Velocity-steered Median Filter V = 2000, 3000 and 5000 m/s |
| Bandpass Filter | 42-48-200-220 Hz |
| AGC | 250 ms |
| Residual Statics | 2 passes |
| Top Mute | Offset-based, 50 ms below the first-breaks |
| Output Data | SEG-Y |

* 3D binning applied for KPreSDM; ** Long-wavelength statics removed before migration.

### 2.5. Basic Theory of Ray-Based and Wave-Equation-Based Depth Imaging Methods

Pre-stack depth imaging methods are in use since the 1970s and had been applied to a wide variety of case studies. With different degrees of structural complexity faced by seismic imaging, different methods were developed and are still evolving, borrowing different concepts from each other. All these methods (either ray-based or wave-equation-based) can be described in terms of wavefield continuation followed by imaging [33]. For example, Kirchhoff migration and beam migration are the most commonly used ray-based methods, whereas wave-equation-based migration methods further divides into two major classes: one-way and two-way. Ray-based methods involve migration of individual (or group of) traces i.e., single arrival (Kirchhoff migration) or multiple arrivals (Gaussian beam migration). They are restricted to smoothly-varying media with small velocity contrasts and where the length scale of heterogeneities is larger than the dominant wavelength. Conversely, wave-equation methods involve the downward continuation of the entire wavefield. Different forms of one-way wave-equation methods based on various approximations and discretization have been proposed over the years. One-way wave-equation migration is believed to produce better results than ray-based methods if multipathing occurs [34]. But the presence of steep dips and complex media undermines the one-way migration techniques and requires two-wave wave-equation-based techniques like RTM for better imaging.

The Kirchhoff pre-stack depth migration method (KPreSDM) is based on the integral solution of the wave equation in the form of the diffraction stack integral [35]. The basic idea behind KPreSDM is that, for a given diffraction point in the subsurface, traveltimes are calculated with respect to each source-receiver pair for a given velocity value. The traveltime information is used to construct a diffraction curve in the form of an ellipse with the source and receiver at the foci. The amplitude associated with the diffraction point is then smeared along this diffraction curve. This process is performed for each point in the subsurface for all source-receiver pairs. If the given velocity field is accurate, constructive interference will produce the image with reflections at their true position. However, with real datasets, the condition of constructive interference is hard to meet especially in cases of reflections emerging from complex geological targets or gaps in shot point coverage typical for hardrock environment. The lack of azimuthal coverage and, generally speaking,



an irregular acquisition is coupled with the difficulties in building a kinematically valid velocity model, which results in sub-optimal imaging with significant noise introduced by the migration operator.

In order to further improve the KPreSDM mainly aimed at better imaging with less noise, another technique called Coherency Migration (CM) was developed [20,23]. Here, instead of a single source-receiver pair, we take advantage of nearby receivers for a given source point. A weighting factor for a given diffraction point is computed as the semblance coefficient ($C^P$), which is defined as the ratio of coherent energy to total energy within a time window over a defined number of nearby traces (see Equation (1)). Here $N$ is the number of neighboring receivers, $T$ is the time window length, $u$ is the wavefield, $t_s$ and $t_r$ are traveltimes from the source and receivers to the image point and P is the exponent factor. As the obtained semblance coefficient is a normalized value, we can use it directly as an additional weighting factor in the Kirchhoff integral. Therefore, for an accurate velocity model and careful selection of nearby receivers, the maximum weightage value will be obtained for an image point at its true location. In addition, intuitively, the obtained image will have suppressed random, incoherent noise. For detailed information on CM, readers are advised to follow [23].

$$C^P = \left( \frac{\int_{-T/2}^{T/2} \left| \sum_{j=0}^{N-1} u_j \left( t_s + t_{r_j} + t \right) \right|^2 dt}{N \int_{-T/2}^{T/2} \left| \sum_{j=0}^{N-1} u_j \left( t_s + t_{r_j} + t \right) \right|^2 dt} \right)^P \tag{1}$$

The third imaging technique is RTM. It is capable of using all types of numerically computable seismic arrivals 'essential for imaging complex media', unlike ray-based methods which are focused on mainly utilizing the primary reflections. RTM is able to distinguish primary reflections from non-primary waves i.e., secondary or multiple reflections produced due to the complexity of the media reducing migration-related artifacts to a great extent [36]. To achieve this, synthetic seismograms are produced by performing forward modeling in the computational domain while the time-reversed observed data is back-propagated independently from each receiver. Both the forward and back-propagated wavefields are cross-correlated at each time step constrained accordingly for a given imaging condition (cross-correlation, interpolated cross-correlation, energy normalized cross-correlation, etc.). A reflector exists where both wavefields are co-located [37]. As a result, RTM produces much more detailed depth images; however, it is more computationally expensive as compared to KPreSDM or CM, and also requires an accurate velocity model. For more details on the developmental history of RTM, please refer to [36].

### 3. Results

This part is divided into two sections; first, the velocity-model building part is briefly covered and, secondly, the results of the depth imaging are presented. First-arrival traveltime tomography (FAT) was used to build the velocity model. We used all the shots and receivers for both lines for depth migration. All the methods are performed within the defined narrow 3D computational domain (red boxes, Figure 1) and in pre-stack mode. However, it should be noted that the KPreSDM was applied to common-offset sections, resulting in Common-Image Gathers (CIGs) ready to be used for velocity-model building or providing quality control of the velocity model, whereas CM and RTM are applied directly to the shot gathers and the migrated output is also in the form of the migrated shot gathers. However, this has the advantage of being able to verify the contributions of individual shots to the image. Same Laplacian filtering [38] was applied as a post-processing filter to all migration results.

### 3.1. Velocity Model Building

For FAT, we used the inversion framework of [39] implemented in the Geotomo Tomo-Plus software. We utilized all the shots and receivers from both Line-A and Line-B to build

the velocity model. Approximately 120,000 traces from Line-A and 128,000 traces from Line-B were semi-automatically picked and manually corrected. A grid spacing of $10 \times 20 \times 5$ m (inline x crossline x depth) was used for traveltime calculations while the inversion was performed with a grid spacing of $20 \times 40 \times 5$ m. Ultimately, a root-mean-squared (RMS) error value of ~5 ms was obtained after 10 iterations. Figures 2b and 3b shows vertical cross-sections along the inline direction of the velocity model for Line-A and Line-B at the same position of corresponding depth images (see red lines in Figures 2a and 3a for inline position). It can be inferred that the meaningful velocity perturbations are observed down to a maximum depth of ~300–400 m for both lines, where the majority of rays reach ~300 m. Below the maximum ray penetration, the model is extrapolated down to a 3 km depth.

### 3.2. Comparison of Migration Results

Kirchhoff migration (KPreSDM) was implemented using the industry-standard TSUNAMI imaging suite (Tsunami Development LLC). The migration was performed in the common-offset planes with output in the form of CIGs. We used the full aperture of the migration operator (90° angle from vertical) and 3000 m distance. An anti-alias filter was applied to improve steeply dipping events. Traveltime calculation and migration were performed with the same parameterization as the input velocity model for both lines, and hence CIGs are produced at $10 \times 20 \times 5$ m (inline x crossline x depth) bins. CIGs were muted and scaled before stacking. Finally, Laplacian filtering was applied to the stacked volume to remove sub-horizontal artifacts and near-surface noise [38]. Figures 2c and 3c shows the result for Line-A and Line-B, respectively. KPreSDM imaged several dipping reflectors for both lines and some near-surface events for Line-B (red arrows, Figure 3c). It failed to image the internal reflectivity in the central area for both lines where reflectivity associated with the Kylylahti formation was expected (black rectangle). The area marked by the black rectangle is only approximate but generally covers the central part of the Kylylahti structure with various rock unit contacts and faults (see cross-section in Figure 1).

The second evaluated method is coherency migration (CM). We used the migration code developed by [23] at the Technical University Freiberg, Germany (TUBAF). As discussed previously, it is an advanced version of KPreSDM where greater focusing of the signal is achieved by taking advantage of the nearby receivers. We tested all the parameters associated with the semblance coefficient (Equation (1)) for optimum imaging, and also to avoid the inception of migration artifacts in terms of over-weighting of the image. CM was also performed on the same migration grid as KPreSDM. We used 21 receivers over a maximum distance of 500 m from the source location and an exponent value of '1' to avoid overweighting the image to calculate the weight for CM for both lines. Migration was performed in the shot domain. Migrated shot gathers were stacked over inline-crossline to produce the migrated volume. Figures 2d and 3d shows the result for Line-A and Line-B, respectively. For both lines, CM achieves a better focusing of events. Several reflections at different depth levels are marked for both lines (see red & black arrows). A distinctive package of reflections in the up-dip directions is also mapped for Line-B (Figure 3d, black arrows between crossline 350–550). Increased reflectivity in the central area (black rectangle) for both lines is also obtained but the reflectors are only piecewise continuous, and it is hard to differentiate between different events. This piecewise continuous reflectivity is a characteristic of the Kylylahti formation as determined from regional 2D seismic profiles [29]. In an overall sense, CM produced less-noisy & better-focused images compared to KPreSDM (compare Figure 2c,d, and Figure 3c,d).

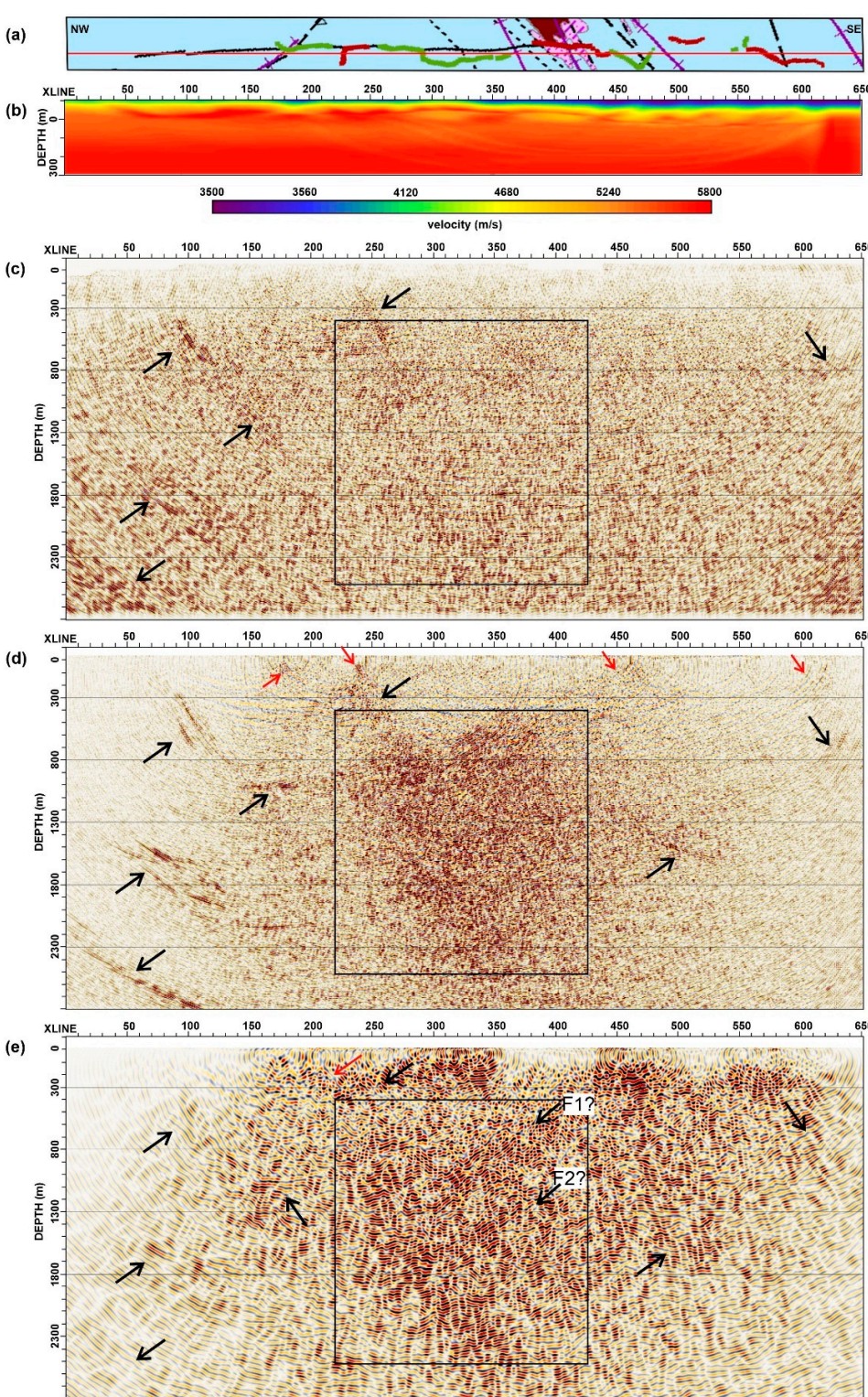

**Figure 2.** Line-A pre-stack depth imaging results. (**a**) Stripe of the geological map restricted to the computational domain (see red box for Line-A in Figure 1), (**b**) velocity model built from FAT (only the top 400 m of the model is shown), (**c**) Kirchhoff migration, (**d**) Coherency migration, and (**e**) RTM. The red line in (**a**) shows the inline location at which the velocity model in (**b**) and depth images in (**c**,**d**) and (**e**) are shown. Black arrows mark some coherent reflectors, red arrows show small-scale near-surface reflectors and the black rectangle marks the central area where higher reflectivity is expected.

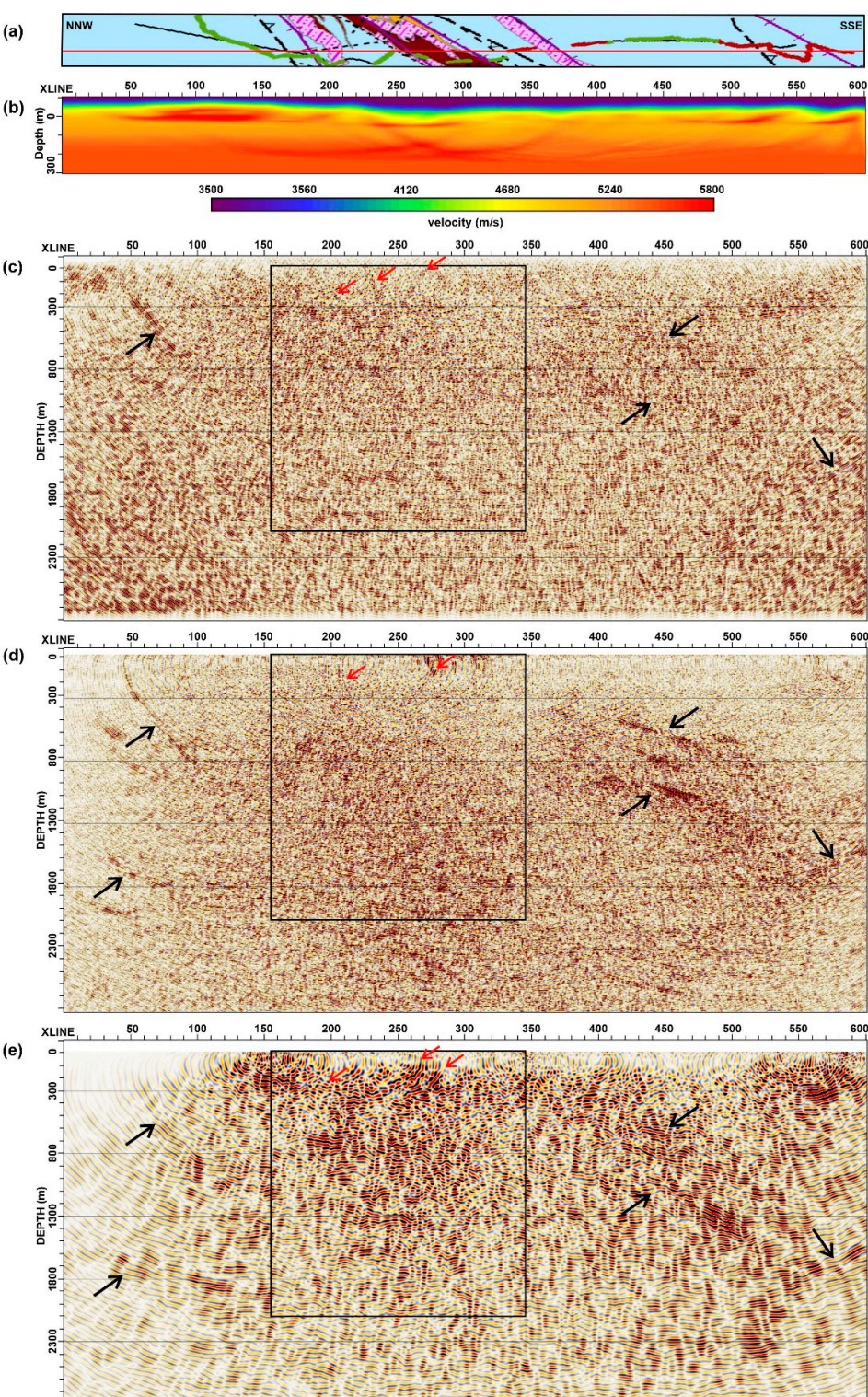

**Figure 3.** Line-B pre-stack depth imaging results. (**a**) Stripe of the geological map restricted to the computational domain (see red box for Line-B in Figure 1), (**b**) velocity model built from FAT (only the top 400 m of the model is shown), (**c**) Kirchhoff migration, (**d**) Coherency migration, and (**e**) RTM. The red line in (**a**) shows the inline location at which the velocity model in (**b**) and depth images in (**c–e**) are shown. Black arrows mark some major coherent reflectors, red arrows show small-scale near-surface reflectors and the black rectangle marks the central area where higher reflectivity is expected.

The third method is RTM. We used RTM code from Reveal software (courtesy: Shearwater Geoservices). We used a 70 Hz minimum-phase Ricker wavelet as source wavelet, convolutional perfectly matched layer boundary condition, offset-dependent migration aperture and 10% aperture taper for suppressing the migration noise at the edges. We set the zero-lag cross-correlation as the imaging condition. As RTM is computationally very intensive, we performed the migration based on the optimum grid size and time step adaptive to the input velocity model. Laplacian filtering was applied to migrated shot gathers and then the final stack was produced by stacking all the shot gathers in the inline-crossline mode. Figures 2e and 3e shows RTM result for Line-A and Line-B, respectively. The obtained images are 'wiggly' in nature and overall somewhat noisy. Several near-surface and deeper events (marked by red and black arrows) are delineated but they are less pronounced compared to CM for both lines. For Line-A, RTM delineates a series of reflections dipping in the NNE direction in the central region (marked as F1 and F2 inside the black rectangle, Figure 2e), whereas for Line-B, a package of nearly-horizontal reflections between the depth range of 600–1800 m are also obtained (black rectangle, Figure 3e).

## 4. Interpretation and Discussion

The main aim of our study was to test the effectiveness of different depth imaging methods applied to crooked-line hardrock seismic data acquired over the challenging Kylylahti structure. Line-A was acquired with the aim of investigating the possible extension of the Outokumpu ultramafic rocks downdip (i.e., southwards), and Line-B to verify whether we are able to image the internal structure of the Kylylahti sequence, as well as the known mineralization. A full geological interpretation of the area at this stage is beyond the scope of this paper. Therefore, our main focus is to validate the different seismic imaging results based on the available geological information. Towards this end, we use Boliden's common-earth model (CEM) built mainly from drillholes and constraints from potential field methods. We primarily use the modeled contact of the black schist (Kalevian metasediments) with the OUM-OME rocks, which can be called an outline of the Kylylahti formation. KPreSDM produced noisy images and was only able to map some major dipping events. In comparison, CM produced a much more focused image with higher reflectivity in the central region but was otherwise piece-wise continuous and difficult to interpret. RTM produced less focused events compared to CM but resulted in superior imaging in the central area for both lines. As the central region is our main area of interest associated with the Kylylahti structure, therefore our interpretation will be based only on the RTM results.

As Line-A is located at the end of the mapped Kylylahti formation, there are no deep drillholes present reaching the depth of our interest (500–2500 m). The CEM model is only reaching the northernmost migrated inline from Line-A. Figure 4a shows the interpretation of the RTM results for Line-A along an inline at 100 m passing almost through the center of the survey. The depth section shows increased reflectivity mapped in the shallow section and the central area. Given the SSW plunge of the Kylylahti formation, an intuitive projection of the reflective updip events (marked as F1 and F2) suggests that they might originate from many of the cross-cutting faults labeled in the vertical section of the Kylylahti formation (top-right panel in Figure 1, also see Figure 2e). Other than many dipping events, there are also large-scale reflectors mapped at different depths (follow the black arrows). A scan through the image cube also reveals a good correlation of the deeper reflective events with the downdip projected modeled contact of black schists with the OME/OUM rocks (dashed purple lines), suggesting a further downward continuation of the Kylylahti formation between the depth range of roughly 2100–2500 m (b.s.l). Figure 4b shows a view from NW for an inline at 120 m and a crossline at 340 m. One of the exploration drillholes is located close to the seismic profile, but it was inclined towards NE. The drillhole intersected a zone of altered rocks (OME) sitting within the Kalevian schists, which seem to correlate with bright reflectors visible at the crossline located at the intersection with the borehole

plane (see the red arrows in Figure 4b). The deeper reflectors (marked by black arrows in Figure 4b) also follow repetitions of OME/OUM and black schists visible in the drillhole log. Thanks to this correlation, we can infer that the reflections visible in the central area of Line-A are characteristics of the rock types forming the Kylylahti formation, which is extending further southwards along the plunge. This example also confirms the utility of the 3D processing/imaging approach applied to crooked line seismic data, even in the case of relatively small crossline aperture. Three-dimensional migration was found beneficial for extrapolating information from the drillholes along the structural dip as seen by 2D profiles [15].

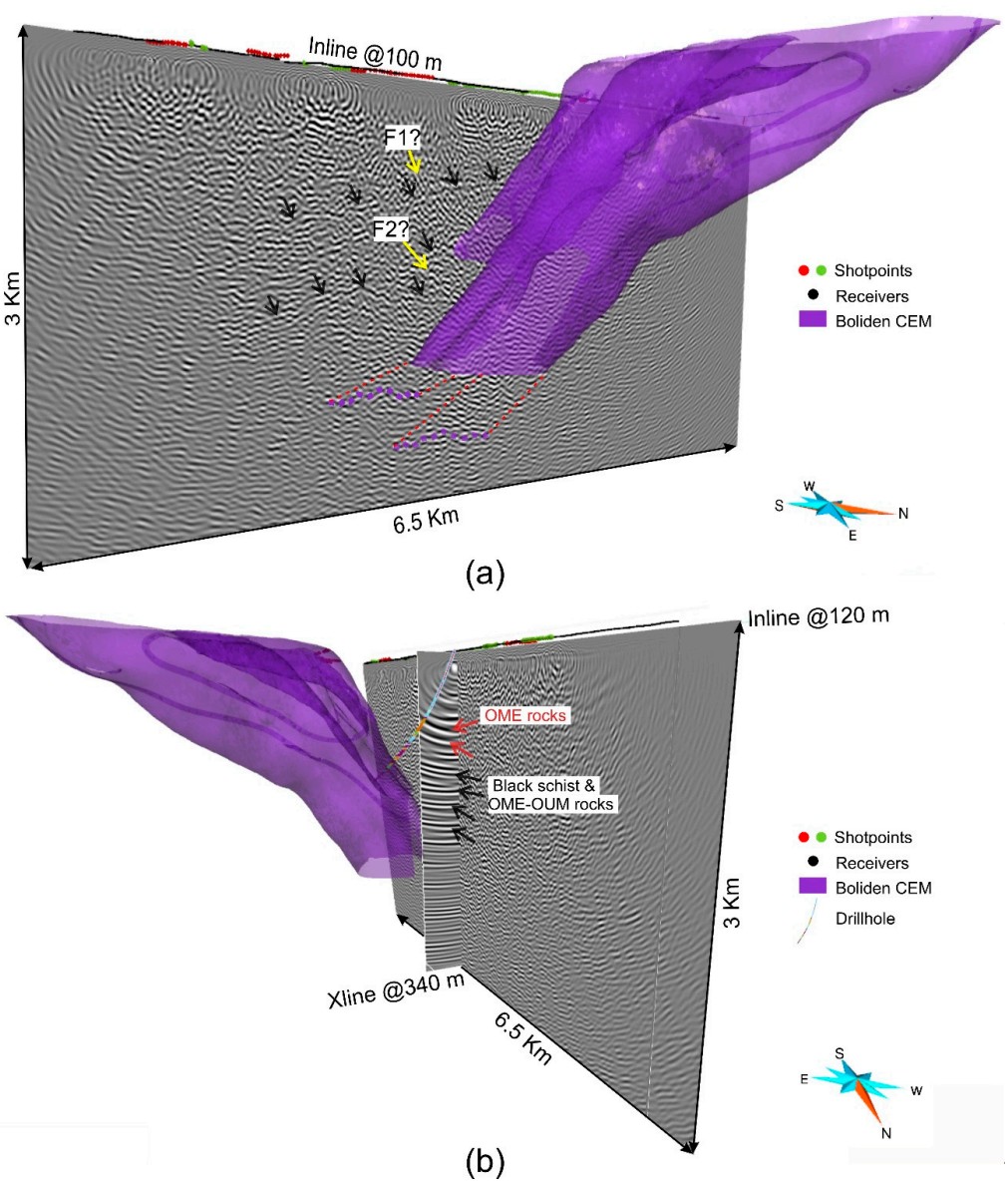

**Figure 4.** Interpretation of RTM results for Line-A. (**a**) shows inline at 100 m distance together with different mapped events (black and yellow arrows) and projection of the black schist—OUM/OME contact (dashed blue lines), (**b**) shows inline at 120 m distance and a crossline at 340 m distance coinciding with the drillhole plane. Red arrows mark the reflections related to the OME rocks sitting within the metasediments. Black arrows point to the reflections likely originating at the repetitions of black schists and OME-OUM rocks, as indicated by the drillhole lithology.

Figure 5 shows the interpretation of the RTM results for Line-B. Figure 5a shows a 3D view of an inline at a 340 m distance with respect to the CEM surface outlining the Kylylahti formation. Figure 5b shows the same inline as in Figure 5a with different features interpreted based on available information. Similar to Line-A, better reflectivity is obtained in the shallow section and central area, and several other reflective events are delineated. Black arrows in Figure 5b point towards a package of reflectors that were independently interpreted as layers of black schist. The solid black lines show fault surfaces already known in the area and dashed blue line shows the intersection of the CEM surface shown in Figure 5a. A good correlation was found between the mapped reflectors and the fault surfaces, validating the accuracy of the RTM image. Red dashed lines show the possible extension of the mapped faults based on the RTM image. This also shows correspondence with the mapped fault surface in the case of Line-A (series of black arrows on top in Figure 4a) suggesting the presence of a major overthrust fault in the Kylylahti area. The reflectivity of the central area also corresponds well with the CEM. The reflections appear geologically plausible and they are contained within the Kylylahti formation. Reflectors marked by red arrows suggest the presence of faults cross-cutting through the Kylylahti formation. We also compared different events observed in the RTM results against the additional apriori knowledge available in the area in Figure 6. Figure 6a,b focus on comparing the RTM results with the CEM model in the shallow region. The mapped reflections are in good agreement with the CEM in general (see black arrows), whereas some events show a similar trend but are distant with respect to CEM (see red arrows). It should be noted that Line-B intersected the actual Kylylahti deposit. On investigation, RTM shows focused reflectivity in the vicinity of the S/MS sulfide lense (see Figure 6c,d). Reflectors likely associated with the mineralization can be tracked across the neighboring inlines for a distance of ~160 m, which provides confidence in such interpretation. It is worth noting that it was only possible to image this reflector through RTM. It is missing in both KPreSDM and CM.

Another important aspect associated with the migration is the computational cost. All results were produced using a mix of commercial software and in-house developed codes; therefore, they exhibit different levels of optimization. Nevertheless, we tried to standardize our computation time based on the same computational server with the exact number of processors (see footnote of Table 3). KPreSDM was the fastest among all methods. CM was significantly slower than KPreSDM. RTM, as expected, was the most expensive among the three methods (see Table 3). However, it should be noted that KPreSDM and CM were performed on the same migration grid, while RTM was adopted for the most optimum grid spacing and time stepping to avoid spatial and temporal dispersion. Therefore, RTM was run using a grid spacing of ~6 m and a time step of ~0.65 ms for both lines as the velocities and other associated processing parameters were similar. Only the final output was produced on the migration grid as defined for KPreSDM and CM.

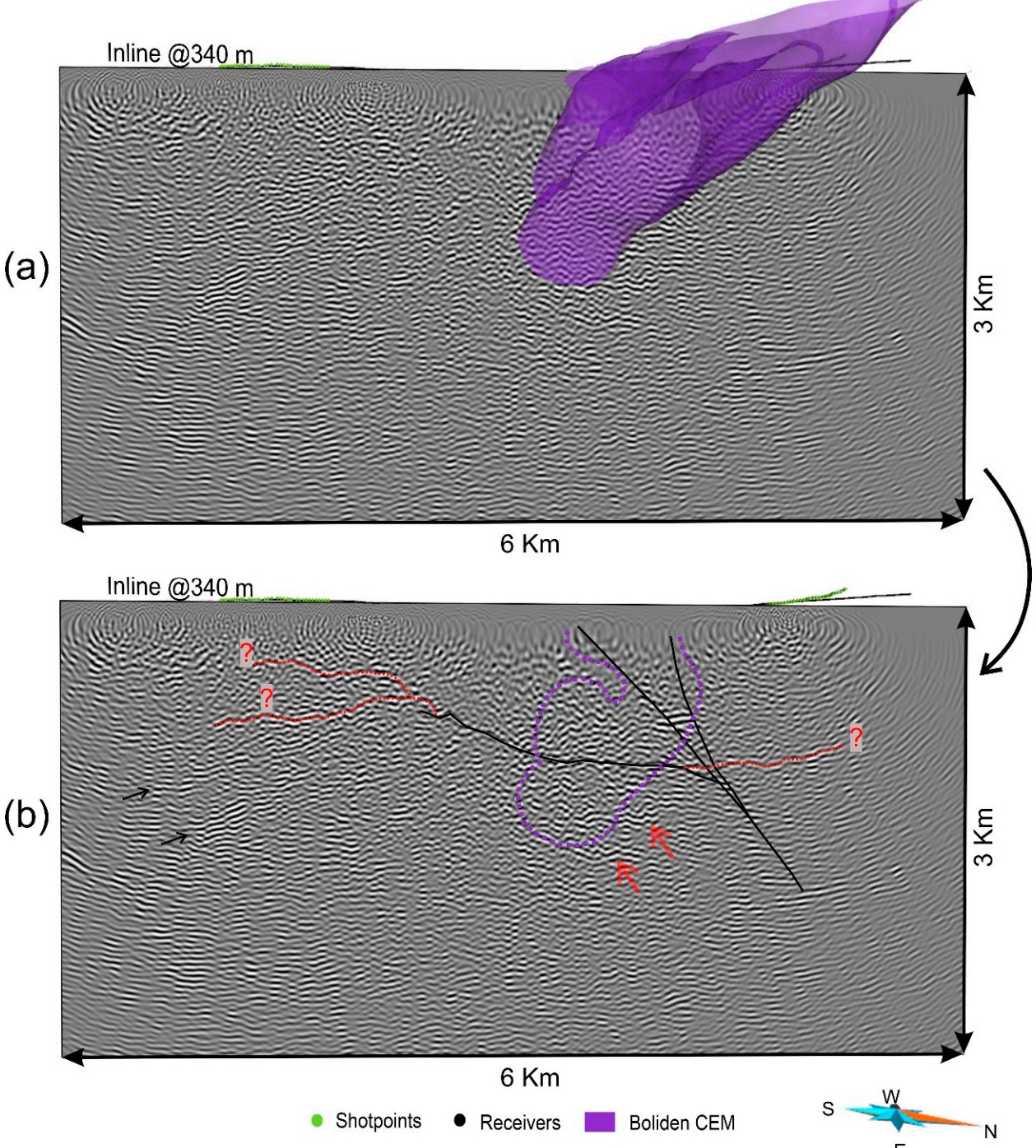

**Figure 5.** Interpretation of RTM results for Line-B. (**a**) 3D view of an inline and the modeled base of the Kylylahti formation (CEM), (**b**) shows the same inline as (**a**) with mapped fault surfaces (solid black lines), their possible continuation (dashed red lines) and the intersection of the CEM surface with the seismic section (dashed blue lines). Red arrows indicate the possible presence of cross-cutting faults through the Kylylahti formation.

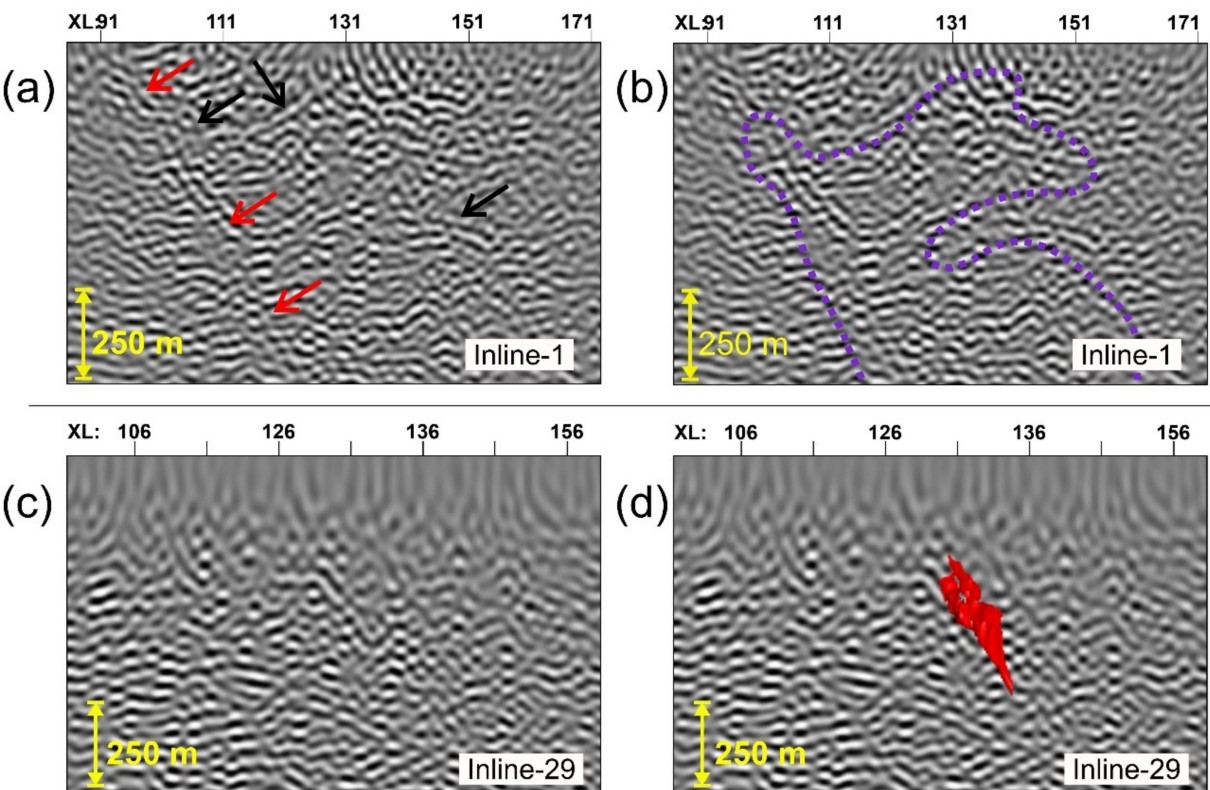

**Figure 6.** Comparison of RTM results for Line-B against apriori knowledge available in the area. (**a**,**b**) zoom on a shallow section, and compare interpreted reflectors (arrows) with the outline of the CEM surface, (**c**,**d**) zoom on the area of the actual intersection of the seismic line and S/MS sulfide mineralization (red surface).

**Table 3.** Computational time for different migration methods for COGITO-MIN 2D profiles.

| Line | KPreSDM | CM | RTM |
|---|---|---|---|
| Line-A | ~1.3 h | ~19.6 h | ~112.8 h |
| Line-B | ~1.5 h | ~29.2 h | ~137.3 h |

Intel(R) Xeon(R) Gold 5120 (using 28 processors).

Despite the higher computational cost, RTM was found superior in terms of imaging the complexity of the geological structure related to the Kylylahti formation and its ability to image the mineralization. Such computation-related challenges can be overcome with more sophisticated implementations e.g., optimizing codes on the accelerated graphical processing units (GPU) [40]. With more regular data acquisition and a much more detailed velocity model derived, e.g., from FWI, a further reduction in the 'noisy' appearance of the RTM sections is expected [25,26]. Conversely, CM seems to be advantageous for mapping the most coherent reflectors. Another advantage of CM compared with RTM is its ability to obtain a well-focused image using only a handful of migrated shot gathers (see the decimation tests in [19]). RTM with a subset of shot points results in a very noisy image. An overall summary of the effectiveness of all the tested methods in mapping different features at various scales is provided in Table 4.

**Table 4.** Comparison of the tested depth imaging methods in mapping different geological features at various scales.

| Feature | KPreSDM | CM | RTM |
|---|:---:|:---:|:---:|
| Steep events close to surface | ✓ | ✓ | ✓ |
| Large-scale structures/reflections | ✓ | ✓ | ✓ |
| Detailed reflectivity inside the Kylylahti structure | X | X | ✓ |
| Reflection from the ore lens | X | X | ✓ |

## 5. Conclusions

We have compared different ray-based and wave-equation-based depth imaging techniques using two crooked 2D seismic profiles acquired in eastern Finland over the Kylylahti deposit. We showcased pre-stack depth imaging results obtained from Kirchhoff migration, coherency migration and reverse-time migration for both lines. Kirchhoff migration produced noisy images and was only able to map major dipping events. Coherency migration was successful in delineating several events at various depths and increased reflectivity within the Kylylahti formation. It produced the most focused and least-noisy images among all methods. RTM produced wiggly and less focused images attributed to the lack of a detailed velocity model. However, it was the only method to map geologically plausible reflectivity in the Kylylahti formation, including direct imaging of the shallow massive sulfide lense. We advocate that in cases of a complex geological setting such as the folded and faulted sub-vertical Kylylahti structure, wave-equation-based migration techniques should be preferred, even though they require high computational resources.

**Author Contributions:** Conceptualization, B.S. and M.M.; Methodology, B.S. and M.M.; Software, B.S.; Validation, B.S. and M.M.; Formal Analysis, B.S. and M.M.; Investigation, B.S. and M.M.; Resources, M.M.; Data Curation, M.M.; Writing–Original Draft Preparation, B.S. and M.M.; Writing—Review and Editing, B.S. and M.M.; Visualization, B.S. and M.M.; Supervision, M.M.; Project Administration, M.M.; Funding Acquisition, M.M. All authors have read and agreed to the published version of the manuscript.

**Funding:** The COGITO-MIN project was funded within the ERA-MIN network. At the national level, the project was supported by Tekes (Business Finland) in Finland and the National Center for Research and Development (NCBR) in Poland.

**Data Availability Statement:** Seismic data can be made available upon request.

**Acknowledgments:** We thank numerous people from the University of Helsinki, the Geological Survey of Finland, IG PAS, Boliden and Geopartner engaged in COGITO-MIN fieldwork. We thank the following vendors for donating the software licenses: Emmerson (GOCAD), IHS Markit (Kingdom Suite) and Shearwater Geoservices (Reveal). Academic licenses of Globe Claritas (Petrosys Ltd.) and TomoPlus (Geotomo LLC) were used for data processing and model building. We especially thank F. Hlousek and S. Buske for providing us with their codes for the specialized Kirchhoff migrations (CM, FVM). Boliden is acknowledged for sharing geological data used to benchmark seismic interpretation. We extend special thanks to two anonymous reviewers for their views which helped in the improvement of the manuscript.

**Conflicts of Interest:** The authors declare no conflict of interest. The funders had no role in the design of the study; in the collection, analyses, or interpretation of data; in the writing of the manuscript; or in the decision to publish the results.

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
