# Peer review of "Seismic Imaging of Mineral Exploration Targets: Evaluation of Ray- vs. Wave-Equation-Based Pre-Stack Depth Migrations for Crooked 2D Profiles"

_minerals, doi:10.3390/min13020264_

Round 1
Reviewer 1 Report
In recent years, efficient evaluation of minerals can be achieved through the use of modern methods of deep seismic survey using reflected waves within the framework of the common depth point technique. Its main advantages are: the multiplicity of observations, the possibility of high-speed analysis, high signal-to-noise ratio. The use of the deep seismic survey technique is especially important in the absence of a lack of sufficient engineering and geological surveys in the fields. In the article, the authors conducted extensive comprehensive studies of depth imaging in fields in Finland. A comparative evaluation of various methods was carried out, and results of interest to readers in the field under consideration were obtained.
However, it would be necessary to clarify a number of comments that are available to the article:
1. In section 2.1. it would be possible to provide a geological map of the considered copper deposits for greater clarity of the given initial data on the geological origin.
2. The article should explain in more detail how Table 2 was obtained, disclose the approaches used for the shortwave and longwave parts, and show the difference between them.
3. From Figure 2 it is not entirely clear how the conclusion is drawn about the area where higher reflectivity is expected (shown by the black box)?
4. This paper should elaborate on a comparative analysis of the validation of the effectiveness of various depth imaging methods applied to curvilinear hard rock seismic data (Section 4). It is possible to provide a comparative analysis of various methods in tabular form.
5. It is interesting to build mathematical models of the results presented in Figures 4 and 5 with the reduction of the reliability coefficients of the available data and predictive data, which is important in the comprehensive assessment of similar deposits.
6. Is it planned to test the obtained results in other fields in the future? Are there any acts of implementation of the obtained results at the production facility, in particular, at the fields in Finland?
7. The list of references could include studies by Chinese and American scientists engaged in research in the area under consideration
Author Response
Brij Singh
PhD Student
Institute of Geophysics
Polish Academy of Sciences
ul. Księcia Janusza 64
01-452, Warsaw
Poland
bsingh@igf.edu.pl
[January 23, 2023 Warsaw]
Dear Reviewer,
Thank you very much for accepting my manuscript for review and providing me with your comments. I have tried my best to incorporate all the valuable suggestions in the revised manuscript. A detailed answer to all the comments can be found in attached files (please see attachment). All the changes in the revised manuscript are in the track-changes mode with the line numbers clearly mentioned in the replies for easiness. I hope all the incorporations fully suffices the issues raised in the current form. I looking forward to the future developments.
Thank you once again for your consideration!
Sincerely,
Brij Singh

Reviewer 2 Report
The document is interesting, my background is on O&G exploration. The main conclusion is well-known for 20+ years in our community, meaning wave-based methods perform better when facing complex geometries and extreme reflectivity ranges, and even before to get empirical evidence just the theory alone yields the same idea, given the assumptions when solving ray-based methods. So the contribution of the work is strictly the test of that hypothesis on this particular case. I would like to see this clearly stated, and maybe add references that support this knowledge from other communities.
In terms of references, I suggest to add:
https://library.seg.org/doi/pdf/10.1190/int-2018-0125.1
Santos, Raphael & Cruz, joão. (2019). Separation of PP and PS converted waves applied in mining exploration. 1-6. 10.22564/16cisbgf2019.233.
@article{eage:/content/journals/10.1111/1365-2478.12895, author = "Papadopoulou, Myrto and Da Col, Federico and Mi, Binbin and Bäckström, Emma and Marsden, Paul and Brodic, Bojan and Malehmir, Alireza and Socco, Laura Valentina", title = "Surface‐wave analysis for static corrections in mineral exploration: A case study from central Sweden", journal= "Geophysical Prospecting", year = "2020", volume = "68", number = "1 - Cost‐Effective and Innovative Mineral Exploration Solutions", pages = "214-231", doi = "https://doi.org/10.1111/1365-2478.12895", url = "https://www.earthdoc.org/content/journals/10.1111/1365-2478.12895", publisher = "European Association of Geoscientists & Engineers"}
In terms of processing, I would like to see sensitivity of the migration results wrt source frequency variation, levels of smoothness of the velocity models and statics pre-processing approaches.
In terms of computing costs, 3D is a must and we are in 2023 so no excuses about computing power, a 3D RTM on GPUs will eat this data in minutes to hours max. Also, you will need this 3D angle for a more detailed VMB by FWI.
M. Araya-Polo et al., "Assessing Accelerator-Based HPC Reverse Time Migration," in IEEE Transactions on Parallel and Distributed Systems, vol. 22, no. 1, pp. 147-162, Jan. 2011, doi: 10.1109/TPDS.2010.144.
and that is 2010!
Author Response

(The authors gave the same response as above.)

Round 2
Reviewer 1 Report
I recommend the corrected article for publication
Reviewer 2 Report
My concerns were mostly address.